# LaTable: Towards Large Tabular Models

## Abstract

Tabular data is one of the most ubiquitous data modalities, yet the literature on tabular generative foundation models is lagging behind its text and vision counterparts. Large Tabular Models (LTMs) could revolutionize the way tabular data is used: not as any single dataset analyzed in a vacuum, but contextualized using their metadata and with respect to related datasets. Creating an LTM is difficult, due to the heterogeneous feature spaces of different tabular datasets, metadata, and prior knowledge. In this work, we propose LaTable: a novel tabular diffusion model that addresses these challenges. We show LaTable can be trained across tabular datasets. Through extensive experiments, we find that LaTable displays early signs of scaling laws previously encountered in foundation model regimes. Moreover, LaTable outperform baselines in out-of-distribution few-shot data generation.

## 1 Introduction

**Motivation.** Foundation models in the image and text domains epitomize the value of large scale training with impressive results that push the boundaries of AI capabilities. In this work, we propose LaTable, a tabular diffusion model that can be trained across vastly different datasets and demonstrates early signs of scaling laws, in hopes that LaTable become the basis for a tabular foundation model.

Tabular foundation models, or Large Tabular Models (LTMs), lag far behind their text and vision counterparts, and existing LTM research focuses on representation and supervised learning Van Breugel & Van Der Schaar (2024). Despite this lag, tabular data is ubiquitous in the real world Borisov et al. (2022a); Shwartz-Ziv & Armon (2022), ranging from electronic health records Fatima & Pasha (2017) to census data Doxsey-Whitfield et al. (2015), from cybersecurity Buczak & Guven (2015) to credit scoring Dastile et al. (2020), and from finance to natural sciences Huang et al. (2012). A generative LTM could transform these fields through enabling few-shot generation of synthetic data, and providing a base model that can be fine-tuned to other tasks like representation learning or prediction van Breugel & van der Schaar (2023). Few-shot capabilities are especially interesting for the tabular domain, where datasets are often smaller, and augmenting existing datasets via few-shot generation of additional columns is of significant interest to researchers and practitioners alike.

**Aim.** Developing a generative LTM is difficult. First, until recently there has been a lack of large tabular metadatasets. Second, different datasets cover vastly different features, and the existing tabular datasets may also use different formats or lack labels for any specific task. Third, tabular datasets may require domain knowledge, or some prior on tabular features. As a result of this, training models meaningfully on large amounts of heterogeneous data remains underexplored. In this paper, we take a step towards LTMs by developing a versatile tabular generative model with the following desiderata:

**D1 Cross-dataset generation.** We need an LTM to be able to generate different tables, which requires it being able to generate different features and variable number of features.

**D2 Categorical and numerical feature generation.** Tables consist of combinations of both continuous and discrete data, and we want to be able to model each explicitly. [1]

**D3 Use textual context.** The model will need to understand contextual metadata, including the dataset description, feature names, and allowed categories for categorical features.

**D4 Equivariance w.r.t. column order.** Table column order is usually arbitrary, which poses an equivariance desiderata. If we define generative model $\mathcal{G} : \mathcal{S}^L \times \mathcal{M} \to \mathcal{O}^L$, where

---

[1] In contrast, an LLM-based approach like Borisov et al. (2022b) implicitly models each numerical feature as a series of discrete tokens. See Section 2 for more details.

$\mathcal{S}$ denotes the column-wise input space (e.g. feature names), $\mathcal{O}$ the output space, and $\mathcal{M}$ the metadata (e.g. dataset description), we have that $\forall s \in \mathcal{S}^L, m \in \mathcal{M}$, we desire $\mathcal{G}(\mathcal{T}(s), m) = \mathcal{T}(\mathcal{G}(s, m))$, where $\mathcal{T}$ represents some permutation to the column order.

**Contributions.** Our contributions are as follows:

1. We introduce LaTable, a novel tabular diffusion model that satisfies these desiderata. Thus, LaTable can be trained across vastly different tabular datasets including different features, number of features, different feature types, and with context of table metadata.

2. We demonstrate that LaTable displays early signs of scaling laws, familiar to those in foundation model regimes. We use both zero-shot and few-shot generation for evaluation.

3. We empirically show that LaTable outperforms existing generative models in out-of-distribution few-shot data generation.

## 2 RELATED WORK

**Tabular single-dataset generative models.** Tabular data is challenging due to its lack of structural meaning (unlike images), mixed-type variables, and often limited size Van Breugel & Van Der Schaar (2024). A growing body of work is developing generative models tailored to tabular data Choi et al. (2017); Xu et al. (2019); Watson et al. (2023), including score-based diffusion models Dieleman et al. (2022); Kotelnikov et al. (2023); Zhang et al. (2023). Nonetheless, most methods ignore or naively one-hot-encode categorical data (e.g. feature names and categories), which loses context that could help overcome data availability issues (**D2 D3**). Naturally, methods developed for single datasets are also not designed for cross-dataset generation (**D1**), or equivariance with respect to columns (**D4**).

**LLM-based approaches.** Pretrained language models may provide a solution to cross-dataset generation, as they contain general knowledge and can represent strings as numerical vectors in a space structured by language supervision. Uniformly, all the works in this direction convert the rows of tables into strings that can be processed by the LLM. The string generated by the LLM is then converted back to a tabular format. Borisov et al. (2022b) fine-tuned GPT-2 Radford et al. (2019) on tables, and they aim to achieve approximate equivariance (**D4**) through randomly generating feature permutations, and sampling token distributions autoregressively rather than sampling only the most likely sentences. Solatorio & Dupriez (2023) note that the approach in Borisov et al. (2022b) retains GPT-2's original vocabulary, yet most tokens may not appear in the tables of interest. They instead use a fixed-set vocabulary Padhi et al. (2021), which reduces the chances of generating invalid samples and thus improves efficiency. Zhao et al. (2023) show that an untrained, smaller LLM can generate more accurate data and do so more cheaply than Borisov et al. (2022b). We note that in contrast to us, previous works like Borisov et al. (2022b); Solatorio & Dupriez (2023); Zhao et al. (2023) fine-tune their model on just a single tabular dataset, i.e. they do not attempt cross-dataset training and do not aim to create a generative model that generalizes beyond the training data (**D1**).

The advantage of LLM-based generation is its simplicity and the fact that it does not require any preprocessing of data, since the raw table can be presented to the LLM directly via prompting and in-context learning. However, adapting LLMs as LTMs comes with serious disadvantages Van Breugel & Van Der Schaar (2024). First, LLMs are expensive during both fine-tuning and inference. A 1-minute training job for CTGAN takes a whopping 540× longer for Borisov et al. (2022b) (see their Appendix B.5, Table 6). One of the core sources of this inefficiency is the linearization of table rows as sentences, and generating these autoregressively. For tabular data, where many columns are typically numerical, this is problematic—it means that a single numerical variable is implicitly modelled as an autoregressive series of categorical variables (e.g. 1.23 is modelled as $1 \rightarrow . \rightarrow 23$). Hence, generating this number with an LLM requires 3 expensive forward calls to the model (plus one for the field separation token). Fitting the whole table in context might also become difficult with datasets containing hundreds of columns. Furthermore, the LLM training objective is not apt at approximating continuous distributions. Hopkins et al. (2023) show that LLMs do not accurately generate simple distributions (e.g. uniform). Van Breugel & Van Der Schaar (2024) visualize how modelling a simple Gaussian with an LLM is non-trivial. By opting for an end-to-end numerical representation of the data and a diffusion model framework, LaTable circumvents these limitations.

# 3 METHOD

**Summary.** We propose LaTable, which satisfies the desiderata **D1-D4**. LaTable uses an encoder-only transformer as backbone for mixed-type diffusion. The input of the model consists of noised feature values (i.e. from the forward noising process), but also the dataset description, column names, conditioning mask (in case of conditional generation), and boolean missingness mask. All metadata strings (e.g. dataset description and feature names) and categorical features are first encoded using a pretrained LLM into a fixed-length embedding. All inputs are subsequently mapped to a common hidden space using separate element-wise MLPs, and added to provide the direct input of the transformer. The output of the transformer is decoded on a feature-by-feature basis, where categorical features are mapped to probabilities over the original features. See Figure 1 for an overview.

## 3.1 SETUP AND NOTATION

We index datasets with $k$, samples with $i$, and features with $j$. [2] Let us assume we have access to a metadataset $\mathcal{D} = \{D^{(k)}\}_{k=1}^{n_D}$, where each dataset $D^{(k)}$ consists of a description, column names, and the data itself. Let $d_k \in \mathbb{N}$ denote the number of columns for dataset $D^{(k)}$. We define the $j$-th feature space for dataset $D^{(k)}$ as $\mathcal{X}_j^{(k)}$. For categorical data, we let this be a finite set of strings, and for numerical columns it may be some subset of $\mathbb{R}$. Recall that we use $\mathcal{S}$ to denote the column-wise input space (e.g. feature names), and $\mathcal{M}$ the metadata (e.g. dataset description). We can thus write $D^{(k)} = (m^{(k)}, s^{(k)}, X^{(k)})$, with $m^{(k)} \in \mathcal{M}$ the dataset description, $s^{(k)} \in S^{d_k}$ feature names, and samples $X^{(k)} \in \Pi_{j=1}^{d_k} X_j^{(k)}$. Finally, let us denote the indices of the categorical features by $\mathcal{I}_{\text{cat}}^{(k)}$.

## 3.2 GENERATIVE MODEL CHOICE

We use a score-based diffusion model Ho et al. (2020); Song & Ermon (2019) for modelling the data distribution. Diffusion models have increased in popularity thanks to their capability to produce samples of higher quality than GANs and VAEs Goodfellow et al. (2020); Kingma (2013). We use the discrete-time formulation with the Denoising Diffusion Implicit Model (DDIM) noise scheduler Song et al. (2020). In the following sections, we will look into each element of LaTable in more detail. We will look at (1) Transformer backbone (**D1**, **D4**); (2) Textual context (**D3**); (3) Mixed-type variables (**D2**), and analyze how each model design choice satisfies desiderata **D1-D4**.

## 3.3 SATISFYING DESIDERATA

### 3.3.1 TRANSFORMER BACKBONE (D1, D4)

We use an encoder-only transformer at the core of LaTable. The transformer backbone is useful for our setting, as it allows variable-length input and output, which allows us to train across datasets (**D1**). We do not use a positional encoding for the input, such that the transformer is equivariant with respect to the input (**D4**). In contrast, a recurrent architecture would not necessarily satisfy this equivariance property. [3] We denote the transformer's hidden dimension by $d_h$.

### 3.3.2 USING CONTEXT (D3)

To make use of context (**D3**), we will encode the category names, column names, and dataset description using a pretrained LLM encoder. Note that the dataset description is the same across all features, but the other metadata are defined on the feature level. By using a pretrained LLM encoder, we avoid having to learn embeddings for descriptions and categories from scratch. It also allows access to textual similarities, even if our tabular data is limited. For example, datasets may contain column "gender" and "sex", and we need not learn from the data alone that these are closely related.

We denote the pretrained LLM by $f_{LLM}$. Text encodings are of a fixed length $d_f$ for any input string $\mathcal{S}$, such that $f_{LLM} : \mathcal{S} \to \mathbb{R}^{d_f}$. Whenever we encode text data, we will denote this with a bar $\bar{m}^{(K)}$:

---

[2]To avoid excessive clutter, we will leave out indices when they are irrelevant or clear from context.

[3]Importantly, removing the positional encoding means the transformer cannot know which column is which. We remove this undesirable symmetry by adding column information to each input element, in the form of a feature name embedding. See Section 3.3.2 for a more thorough discussion.

1. Each dataset description $m^{(K)}$ is encoded by the LLM into a vector $\bar{m}^{(K)}$.

2. Each categorical variable is converted into a string "[column name s] is [category c]". Categories $(c_1, ..., c_n)$ are then encoded by the LLM and stacked into an embedding matrix $\bar{C}_j^{(k)} = (\bar{c}_1, ..., \bar{c}_n)$.

3. Feature names $s_j^{(k)}$ are encoded by the LLM into vectors $\bar{s}_j^{(k)}$.

We condition on the encoded metadata by mapping the dataset description $\bar{m}$ and feature embeddings $\bar{s}_j$ to size $d_h$ using shallow MLPs $g_r$, $g_s$ and adding it to the transformer input (see Figure 1). For the encoder $f_{LLM}$, we use the model $WhereIsAI/UAE - Large - V1$ for three reasons: it is open source and easily available [4]; it achieved SOTA on the Massive Text Embedding Benchmark (MTEB) Muennighoff et al. (2022); and it is very lightweight (size: 1.34 GB). All encodings are cached to disk before training and testing to avoid LLM forward calls during training.

### 3.3.3   MIXED-TYPE VARIABLES (D2)

In contrast to LLMs, transformers, and diffusion models, we need to generate numerical values as well as categoricals (**D2**). Let us start with the numerical variables.

**Numerical variables.** For numerical variables we can use relatively standard diffusion. Given some time step $t$, the forward denoising process is applied to all numerical features $x_j$ , giving the noisy $(x_j)_t$. A shallow MLP $g_{num}^{in} : \mathbb{R} \to \mathbb{R}^{d_h}$ is used to map each numerical variable to the transformer's input space. The transformer is applied to the sequence of all variables. Subsequently, each sequence element in the transformer's output corresponding to a numerical variable is mapped back to scalars using output network $g_{num}^{out} : \mathbb{R}^{d_h} \to \mathbb{R}$. For each numerical $x_j$, this yields output $\hat{x}_j$, and the loss (e.g. sample-loss, $\epsilon$-loss, or v-prediction) is computed per feature.

**Categorical variables.** To handle categorical variables, we draw inspiration from Continuous Diffusion for Categorical Data (CDCD) Dieleman et al. (2022). The original CDCD learns high-dimensional embeddings for each category separately, before applying $L_2$ normalization to each embedding (after which noise is added for the denoising). During training, CDCD adds noise to these categorical embeddings (identical to continuous diffusion), and trains a model that predicts probabilities for the original categories. This model is trained using some classification loss, e.g. cross-entropy. During sampling, the outputted probabilities are used to predict the score in the continuous space. Specifically, they use score interpolation—which is effectively a weighted mean of all category embeddings, where weights are given by the probabilities predicted by the score model.

In our case, we do not want to learn embeddings and predict probabilities for all possible categories in all datasets. By learning an embedding separately for each category, we lose context that is captured in the category name (e.g. how categories female and woman are related). It also makes it difficult to scale to large numbers of categories, especially if some of these categories are hardly ever observed. Instead, we use the pretrained LLM $f_{LLM}$ for acquiring $d_f$-dimensional embeddings for each category $c$, which carry the contextual knowledge of the category name. [5] We fine-tune these embeddings using a shallow MLP followed by an $L_2$ normalization layer, which we together denote by $g_f : \mathbb{R}^{d_f} \to \mathbb{S}^{d_f}$. [6] The forward diffusion process corrupts these vectors by adding Gaussian noise, giving $\bar{c}_t$. Similar to the numericals, we use a network $g_{cat}^{in} : \mathbb{R}^{d_f} \to \mathbb{R}^{d_h}$ to map these noisy embeddings to the transformer's input space, and network $g_{cat}^{out} : \mathbb{R}^{d_h} \to \mathbb{R}^{d_f}$ is applied to the categorical sequence elements in the transformer's output—giving predicted embeddings in $\mathbb{R}^{d_f}$.

One key question remains: considering each categorical feature may have a different number of categories, how do we map the transformer output (in $R^{d_f}$) back to the individual categories? We achieve this using an attention-like layer. Letting $\hat{c}_j \in \mathbb{R}^{d_f}$ be the output of the model, we let the final predicted probability for some category $i$ be proportional to the similarity of the generated vector and the original categories' embeddings:

---

[4]https://huggingface.co/WhereIsAI/UAE-Large-V1

[5]As discussed in Section 3.3.2, for all categorical features we embed the categorical $c$ as $\bar{c}$ = "[feature name] is [c]".

[6]Without the normalization layer, it would be difficult for the model to give embedded categories with small $L_2$-norm a high probability (see Eq. 1). We add the shallow MLP's finetuning to avoid projecting embeddings that are quite different in $\mathbb{R}^d$ to the same point on the unit sphere.

$$p(c_i) = \text{softmax}(g_f(\bar{C})^T \hat{c})_i, \forall i \tag{1}$$

thereby allowing us to predict probabilities for each datasets' and categoricals' categories individually. Identical to CDCD, we use cross-entropy loss (CE) for the categorical variables and use score interpolation. Combining the numerical and categorical losses, we get:

$$\mathcal{L}(x, \hat{x}) = \frac{1}{d_k}[\sum_{j \in \mathcal{I}_{\text{cat}}^{(k)}} CE(x_j, \hat{x}_j) + \sum_{j \notin \mathcal{I}_{\text{cat}}^{(k)}} MSE(x_j, \hat{x}_j)] \tag{2}$$

**Bringing it together.** Overall, the model thus consists of a number of independent MLPs $g$ that allow us to bring differently sized variables (dataset description, feature column name embeddings, categorical features, numerical features) all to the same space $d_h$. The input of the transformer consists of a sequence of inputs $h_j \in \mathbb{R}^{d_h}, j = 1, ..., d_k$ (recall that $d_k$ is the number of columns in the table), with $h_j = g_{cat}^{in}((\bar{x}_j)_t) + g_r(\bar{m}) + g_s(\bar{s}_j) + g_t(t)$ for categoricals, and $h_j = g_{num}^{in}((x_j)_t) + g_r(\bar{m}) + g_s(\bar{s}_j) + g_t(t)$ for numericals. This way, the encoder-only transformer backbone allows variable-length input and output, which allows us to train across datasets (**D1**). We do not use a positional encoding for the input, such that the transformer is equivariant with respect to the input (**D4**). And We encode all metadata, including dataset description, feature column names, etc using a pre-trained LLM (**D3**). The transformers output sequence is again decoded using independent shallow networks $g_{num}^{out}$ and $g_{cat}^{out}$ to provide score estimates for both numerical and categorical variables (**D2**). See Figure 1 for a full overview.

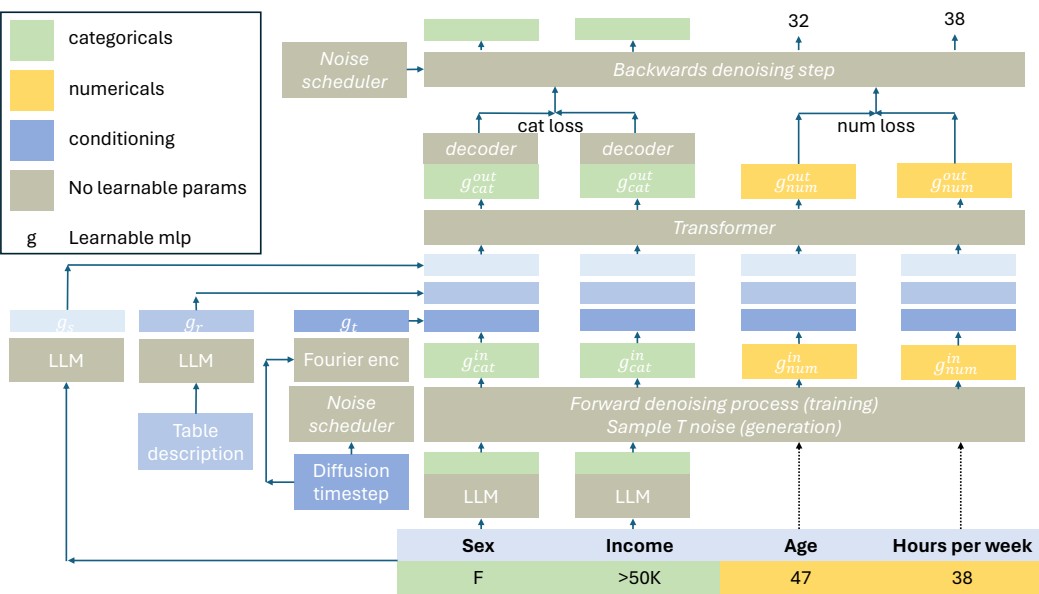

Figure 1: The diffusion pipeline, for both training and generation. Note that the whole pipeline is flexible with respect to the number of numerical and categorical features as input. The LLM encoder is frozen and the transformer is an encoder-only model without positional encodings or causal masking. Additional conditioning (e.g. missingness mask, conditioning information) can be trivially added to the transformer input, or through cross-attention layers.

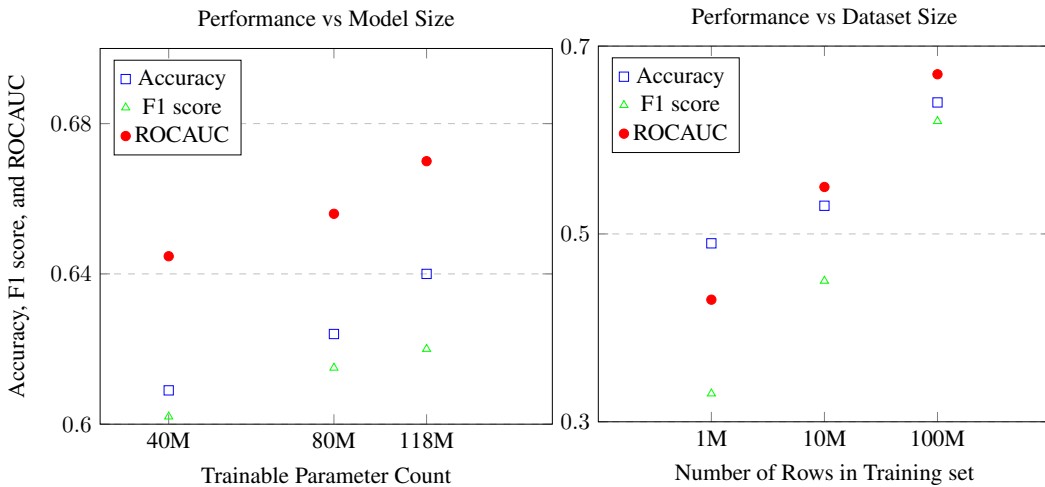

Figure 2: By varying model size and dataset size, we observe that Latable's performance follows a general log-linear trend, mirroring that of early scaling laws in foundation models regimes. That is, as we increase the model/dataset size exponentially, we obtain a linear increase in the final performance.

## 4 EXPERIMENTS

In this section, we will explore the tabular data generation capabilities of LaTable. We show that LaTable displays early signs of scaling laws commonly found in foundation model regimes. Then, we benchmark the performance of LaTable against popular baseline tabular models, as well as ablation experiments that justify the design choice of our method.

**Evaluation Pipeline** We follow a four-step process to evaluate the performance of LaTable in the few-shot out-of-distribution generation setting, partially inspired by the "Train on Synthetic, Test on Real" methodology. (1) We obtain the final model checkpoint after training has converged (under different training settings). (2) We choose a test dataset. More details about how the test datasets are selected and what test datasets we use will be discussed shortly. (3) We perform minor fine-tuning on 100 data samples (which is very small for synthetic data generation models), and we generate synthetic data given the dataset metadata. (4) Finally, we use a well-known classifier model to fit the generated synthetic data, and evaluate on real data using downstream performance metrics such as accuracy, f1 score, and Area Under the Receiver Operating Characteristic Curve (rocauc). By default, we will use the CatBoost classifier because it is has strong performance on tabular dataset tasks.

**Training data.** In order to make meaningful advancement towards foundation models, we need to gather a training set for tabular data whose scale surpasses any existing collections of tabular datasets. We build upon the recent effort of Tablib Eggert et al. (2023), which scrapes over 600M tables from the web. After filtering through the data via heuristics and standard data cleaning, we arrive at a subset of 100K tabular datasets. Finally, we chop each dataset down to 1K rows ensure an even spread of data during training. In the end, the training set totals 100M rows, spanning 30K columns.

**Test data, and combating data leakage.** Because the datasets from Tablib are scraped from the web, there is no sensible way to rule out data contamination or data leakage in the training data. In order to best prevent data leakage and contamination, we gathered several tabular datasets that were not commonly found on the web. These datasets are Cardio, URL, WiDS, Insurance, and Heloc Stoian et al. (2024). In addition, for the tabular datasets that we ask LaTable to generate, we deliberately pick the ones that LaTable cannot generate from zero-shot, hence verifying that LaTable did not remember the training data. The core component of our evaluation is to show that training across a large number of datasets indeed gives LaTable the ability to generate synthetic tabular datasets with very few fine-tuning examples, and potentially pave a path for building larger and better LTMs for future research.

## 4.1 SCALING LAWS: MODEL AND DATA

In this section, we show that LaTable displays early signs of scaling laws commonly found in foundation model regimes. We vary different training settings, and record the final downstream performance from the synthetic data generation after the training has converged. We change the model size by varying the number of transformer layers in the architecture, arriving at model sizes of 40M (2 layers), 80M (6 layers), and 118M (10 layers). We change the dataset sizes by varying the number of tables we include in the training set, arriving at dataset sizes of 1M (1K tables), 10M (10K tables), and 100M (100K tables). When varying the model sizes, the dataset size is fixed at 100M (100K tables). When varying the dataset sizes, the model size is fixed at 118M (10 layers).

As we can see in Figure 2, the performance of LaTable follows a log-linear trend, mirroring that of early scaling laws in foundation models regimes. That is, as we increase the model/dataset size exponentially, we obtain a linear increase in the final performance. Interestingly, the model seems to yield the best performance when trained with roughly the same dataset size as parameter count, which could suggest further directions for tabular scaling laws.

## 4.2 LATABLE OUTPERFORMS BASELINES IN DATA GENERATION

In this section, we show that LaTable outperforms various baseline methods in few-shot out-of-distribution tabular data generation. We choose GREAT, CTGAN, TVAE Xu et al. (2019), ARF Watson et al. (2023), and TabDDPM Kotelnikov et al. (2023) as baselines, all of which are competitive methods for tabular data tasks. We evaluate these methods on the test datasets Cardio, URL, WiDS, Insurance, and Heloc. More specifically, we fit a CatBoost classifier on the generated data (treating the synthetic data as the training data), and obtain downstream performance metrics (accuracy, F1 score, ROCAUC) from using the real data as the test set. In order to generate sensible synthetic data, all methods are first fine-tuned for 100 samples on the test dataset.

Table 1: LaTable outperforms baselines in the few-shot out-of-distribution data generation setting. Because we only fine-tune on 100 samples, baseline methods fail to generate high-quality synthetic data, while LaTable generates good synthetic data, sometimes even coming close to real data. We attribute this to both the training procedure and model design choices, as we will discuss in later sections.

| Methods | Cardio | URL | WiDS | Insurance | Heloc |
|---|---|---|---|---|---|
| CTGAN | | | | | |
|     Accuracy | 0.50 (0.00) | 0.59 (0.01) | 0.55 (0.00) | 0.56 (0.01) | 0.48 (0.01) |
|     F1 score | 0.43 (0.00) | 0.58 (0.00) | 0.54 (0.01) | 0.53 (0.00) | 0.45 (0.01) |
|     ROCAUC | 0.53 (0.07) | 0.60 (0.00) | 0.57 (0.00) | 0.57 (0.00) | 0.48 (0.02) |
| TVAE | | | | | |
|     Accuracy | 0.58 (0.00) | 0.61 (0.00) | 0.57 (0.01) | 0.61 (0.00) | 0.59 (0.00) |
|     F1 score | 0.58 (0.00) | 0.60 (0.02) | 0.56 (0.01) | 0.60 (0.00) | 0.58 (0.00) |
|     ROCAUC | 0.60 (0.01) | 0.64 (0.01) | 0.60 (0.02) | 0.65 (0.01) | 0.60 (0.00) |
| ARF | | | | | |
|     Accuracy | 0.55 (0.01) | 0.53 (0.00) | 0.55 (0.00) | 0.54 (0.00) | 0.53 (0.02) |
|     F1 score | 0.53 (0.01) | 0.52 (0.00) | 0.53 (0.01) | 0.52 (0.01) | 0.53 (0.01) |
|     ROCAUC | 0.57 (0.00) | 0.55 (0.01) | 0.55 (0.00) | 0.56 (0.00) | 0.53 (0.01) |
| DDPM | | | | | |
|     Accuracy | 0.59 (0.01) | 0.66 (0.00) | 0.55 (0.00) | 0.58 (0.00) | 0.60 (0.00) |
|     F1 score | 0.58 (0.00) | 0.65 (0.00) | 0.54 (0.00) | 0.58 (0.00) | 0.58 (0.00) |
|     ROCAUC | 0.59 (0.00) | 0.70 (0.00) | 0.58 (0.00) | 0.62 (0.01) | 0.64 (0.02) |
| GREAT | | | | | |
|     Accuracy | 0.58 (0.00) | 0.59 (0.00) | 0.57 (0.01) | 0.58 (0.00) | 0.57 (0.00) |
|     F1 score | 0.57 (0.00) | 0.57 (0.01) | 0.55 (0.00) | 0.57 (0.00) | 0.57 (0.00) |
|     ROCAUC | 0.58 (0.00) | 0.59 (0.00) | 0.59 (0.00) | 0.57 (0.00) | 0.57 (0.02) |
| **LaTable** | | | | | |
|     Accuracy | 0.64 (0.02) | 0.79 (0.01) | 0.75 (0.01) | 0.81 (0.02) | 0.71 (0.00) |
|     F1 score | 0.62 (0.03) | 0.79 (0.01) | 0.74 (0.00) | 0.80 (0.00) | 0.71 (0.00) |
|     ROCAUC | 0.67 (0.01) | 0.85 (0.01) | 0.80 (0.00) | 0.87 (0.01) | 0.77 (0.00) |

Because we are only fine-tuning the methods on 100 samples (which is very small for tabular data generation tasks), none of the baselines perform particularly well for the downstream metrics, see Table 1. On the other hand, LaTable outperforms all baseline methods in the few-shot out-of-distribution data generation domain. Another performance assessment we can obtain is the ground-truth downstream performance, because we can break down the real data into the training set, validation set, and test set. For instance, when we fit CatBoost on the training set of Cardio, and then evaluated that performance, the accuracy and F1 score comes close to 0.74. This is inspiring, because it suggests that we are able to generate data that is approaching the quality of the read dataset. In future sections, we will discuss why we have reason to believe the architectural design choices made in LaTable contributes to these results.

## 4.3    THE IMPORTANCE OF CROSS-DATASET TRAINING (D1)

Table 2: The training phase is essential for LaTable's good performance. Without training, LaTable's performance drops to baseline level.

| Methods | Cardio | URL | WiDS | Insurance | Heloc |
|---|---|---|---|---|---|
| Without training | | | | | |
|     Accuracy | 0.50 (0.00) | 0.50 (0.00) | 0.49 (0.01) | 0.57 (0.01) | 0.52 (0.01) |
|     F1 score | 0.33 (0.00) | 0.48 (0.01) | 0.37 (0.01) | 0.54 (0.00) | 0.42 (0.01) |
|     ROCAUC | 0.58 (0.06) | 0.53 (0.00) | 0.53 (0.00) | 0.60 (0.00) | 0.58 (0.00) |
| With training | | | | | |
|     Accuracy | 0.64 (0.02) | 0.79 (0.01) | 0.75 (0.01) | 0.81 (0.02) | 0.71 (0.00) |
|     F1 score | 0.62 (0.03) | 0.79 (0.01) | 0.74 (0.00) | 0.80 (0.00) | 0.71 (0.00) |
|     ROCAUC | 0.67 (0.01) | 0.85 (0.01) | 0.80 (0.00) | 0.87 (0.01) | 0.77 (0.00) |

In this section, we will discuss the first design choice (**D1**), and specifically the importance of training LaTable on large amounts of data. We take the first checkpoint and the last checkpoint of LaTable, such that we can benchmark the performance with and without the training phase. As we see in Table 2, LaTable's performance drops significantly when training is not applied before synthetic data generation, and the downstream performance drops to the level of other models as we expect it would. Recalling Figure 2, the model reaches its best performance when it is trained on large amounts of training data. In conclusion, we believe that the good performance of LaTable is not only due to the model architecture, but also the cross-dataset training procedure.

## 4.4    TEXTUAL CONTEXT IS ESSENTIAL FOR PERFORMANCE (D3)

Table 3: LaTable's performance depends on the quality of the metadata it gets. When we switch the default LaTable embeddings (UAE) to dummy embeddings, train until convergence, fine-tune on 100 samples, and generate synthetic data, LaTable's performance drops to baseline level.

| Methods | Cardio | URL | WiDS | Insurance | Heloc |
|---|---|---|---|---|---|
| Dummy Embeddings | | | | | |
|     Accuracy | 0.53 (0.01) | 0.52 (0.01) | 0.54 (0.00) | 0.50 (0.01) | 0.49 (0.01) |
|     F1 score | 0.46 (0.01) | 0.48 (0.00) | 0.47 (0.00) | 0.43 (0.00) | 0.46 (0.00) |
|     ROCAUC | 0.55 (0.01) | 0.54 (0.00) | 0.57 (0.00) | 0.54 (0.01) | 0.53 (0.00) |
| LaTable Embeddings | | | | | |
|     Accuracy | 0.64 (0.02) | 0.79 (0.01) | 0.75 (0.01) | 0.81 (0.02) | 0.71 (0.00) |
|     F1 score | 0.62 (0.03) | 0.79 (0.01) | 0.74 (0.00) | 0.80 (0.00) | 0.71 (0.00) |
|     ROCAUC | 0.67 (0.01) | 0.85 (0.01) | 0.80 (0.00) | 0.87 (0.01) | 0.77 (0.00) |

In this section, we discuss the third design choice (**D3**). Specifically, we ablate on the embedding model used for the metadata, thus capturing the importance of textual input (the embedding model is responsible for dataset description, column names, and other textual data inputs). As we can see in Table 3, the performance of LaTable takes a hit when we replace the UAE embedding model with dummy embeddings, possibly impacting not only training but also the data generation process during evaluation.

## 4.5 FINE-TUNING IMPACT ON FINAL PERFORMANCE

Table 4: The number of fine-tuning samples impacts LaTable's performance. When we increase the number of fine-tuning examples from 100 to 1000, LaTable performs better in data generation, even coming close to the downstream performance of real data.

| Methods | Cardio | URL | WiDS | Insurance | Heloc |
|---|---|---|---|---|---|
| Fine-tune 100 sample | | | | | |
|    Accuracy | 0.55 (0.00) | 0.60 (0.00) | 0.57 (0.00) | 0.60 (0.00) | 0.57 (0.01) |
|    F1 score | 0.53 (0.00) | 0.58 (0.01) | 0.56 (0.00) | 0.57 (0.00) | 0.53 (0.00) |
|    ROCAUC | 0.55 (0.00) | 0.64 (0.00) | 0.58 (0.001) | 0.64 (0.00) | 0.58 (0.00) |
| Fine-tune 1000 sample | | | | | |
|    Accuracy | 0.69 (0.01) | 0.81 (0.00) | 0.79 (0.01) | 0.83 (0.02) | 0.73 (0.00) |
|    F1 score | 0.69 (0.01) | 0.80 (0.00) | 0.78 (0.00) | 0.81 (0.01) | 0.73 (0.00) |
|    ROCAUC | 0.76 (0.00) | 0.86 (0.01) | 0.81 (0.00) | 0.87 (0.00) | 0.80 (0.01) |

In this section, we conclude the experiments with an ablation on the number of samples used for fine-tuning. We change the number of fine-tuning samples from the default (100) to 1000, thereby enhancing the level of access LaTable has to the test dataset during the data generation phase. As we can see from Table 4, the downstream performance increases, coming closer and closer to the real data performance (around 0.72 for Cardio).

## 5 DISCUSSION

**Summary** We have motivated this paper by describing the effect Large Tabular Models (LTMs) could have on the way tabular data is used: not as any single dataset analyzed in a vacuum, but contextualized using their metadata and with respect to related datasets. We acknowledge the difficulty in creating an LTM, due to the heterogeneous feature spaces of different tabular datasets, different specifications of tabular data tasks, and even messy/insufficient data. In order to make a meaningful first step towards creating LTMs, we propose LaTable, which can be trained across tabular datasets (**D1**), takes information from categorical, numerical, and textual features (**D2, D3**), is equivariant with respect to the input (**D4**), and displays early signs of scaling laws previously encountered in foundation model regimes as well as outperform baselines in out-of-distribution few-shot data generation. We would like to conclude this paper with discussions about limitations, impacts, and next steps.

**Binary classification, Multi-class, and Regression** It is insightful to discuss a bit more about the way we select the test datasets. We have selected five test datasets—Cardio, URL, WiDS, Insurance, and Heloc—so that we can best ensure there will not be any data leakage issues. At the same time, all five datasets belong to the same task—binary classification. This is not a coincidence. Intuitively, binary classification is much easier than multi-class classification and regression, for downstream models and synthetic data generation alike. This is because the number of modes for a binary classification data distribution is likely to be significantly lower than other tabular dataset tasks. When running experiments for LaTable, we have attempted to execute the same evaluation pipeline for multi-class classification and regression datasets (like Gesture), but the performance did not come close to the downstream results obtained on binary classification tasks, or the ground truth downstream performance. We postulate that multi-class classification datasets and regression datasets are inherently more complex datasets, and therefore more difficult to generate. This marks the proper understanding of different types of tabular datasets to be a valuable and worthy direction of future research.

**Going forward.** We think it would be beneficial for the community to list out a few directions of LTM research

- **Scaling Laws.** Unlike in text and in vision, we don't have a good intuition of the type of scaling laws that would appear in LTMs. For instance, we do not yet know whether there is a scaling law for model size and dataset size, like the ones we have seen in large language model regimes. Nor do we know the optimal compute ratio when it comes to training and

evaluating LTMs. The proper scale of an LTM model and its dataset is likewise uncertain. All of these are valuable directions for future research.

- **Dataset Tasks.** As we have noted in earlier sections, LaTable performs well across binary classification datasets, but the reason why its performance drops on multi-class and regression datasets still elude us. It is possible that the latter datasets are inherently more difficult to capture. Whether or not that is the case, much more research effort should be spent into understanding the different types of tabular datasets, and how they are different in the nature of their tasks.

- **High-quality data.** Apart from Tablib, we have also tried training on WikiTables, another large-scale tabular dataset in the wild. However, LaTable's performance does not improve with WikiTable training, possibly due to the fact that tables that cover a wide range of topics with context from the Wikipedia article are not high-quality tabular datasets. These experiment results emphasize the importance of good quality data in improving generalization performance of LTMs. The issue of data quality may be resolved by better understanding of large metadatasets like TabLib, and using heavy-handed curation and selection. To avoid data leakage between the training and evaluation sets, this also requires robust tools for identifying and removing duplicates, which is hard for tabular datasets since pre-processed and filtered copies are common. Another possible avenue for future work is to use more specialized databases for training, e.g. of high-throughput biological data.

**Limitations.** LaTable can be extended and improved in multiple ways:

- **Scale and data.** As discussed in the previous section, LaTable is a relatively small model trained on a relatively small dataset. Scaling up LaTable and discovering larger scaling laws is one of the key challenges for future research. In addition, we must simultaneously search for high-quality data that spans binary classification, multi-class, and regression problems.

- **Extending variable and table types.** We restrict ourselves to numerical and categorical data, but future work could consider date time, full string descriptions, time-series, and relational databases to increase applicability and impact. In addition, future work could also explore in-depth the inter-feature relationships and interactions between features as source of information.

- **Bias.** In this work, we did not explore possible bias in the data and pre-trained LLMs, or how LaTable copies that bias. More research into bias is required before applying LaTable to the real-world, see broader impact below.

**Broader Impact.** It is insightful to recognize why LTMs can be difficult to create. Even though there is an abundance of tabular data in the real world, in-the-wild tabular data can also be very large and complex, often containing thousands of columns and millions of rows. This introduces obvious problems such as complicated data preprocessing pipelines, expensive computation, etc. Because the scale of tabular data is so large, even the features that are observed during training might not be fully captured by the model—there may be a significant shift in the feature distribution from train to test, let alone how one feature interacts (e.g. correlates) with other features, and the interaction of features will often require domain knowledge to effectively annotate, i.e. LLMs may not be able to approximate. All the difficulties we have discussed in this paper should be taken into consideration when quantifying the difficulty of creating foundation models for tabular data. Nevertheless, LTMs possess the huge potential to transform how we process real-world data as we know it. We believe the broader impact of LaTable, and continued work in this area, is primarily positive. LaTable can enable better synthetic data using fewer samples, which could promote more responsible AI. For example, to improve minority representation using data augmentation, ML model testing through data simulation, and data democratization through private synthetic data. The latter is especially interesting, since few-shot generation may require a smaller privacy budget than standard synthetic data (after all the model need not learn the full distribution from the private data and may be less likely to memorize this data). Nonetheless, we need to acknowledge the risk: LaTable may make errors and may reflect or exacerbate societal biases that are present in the data, or in the pretrained LLM embeddings. More research into possible biases is required. The quality and fairness of generated data should always be evaluated before applying LaTable to real-world sensitive settings like healthcare and finance. We look forward to future research on tabular foundation models.

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

## A  PSEUDO-CODE

**Algorithm 1:** Training phase. $\Theta$ denotes all learnable parameters (of denoising and primary networks). We use the CDCD noise scheduler(Dieleman et al., 2022) for categoricals and VESDE Song et al. (2020) for numericals, denoted by respectively "CatSched" and "NumSched". Capitals $X$ and $H$ denote the tensors of dimensions (batch size, number of columns, hidden dimension)

---

**Input:** data $\mathcal{D}$, LLM encoder $f_{LLM} : \mathcal{S} \to \mathbb{R}^{d_f}$, diffusion noise schedulers CatSched and NumSched, batch size "bs"

**Initialise:** transformer $g_T$, primary networks $g_f, g_s, g_r, g_t, g_{cat}^{in}, g_{num}^{in}, g_{cat}^{out}, g_{num}^{out}$, cache dictionary $\bar{\mathcal{D}} = \emptyset$.

*# Encode all strings and cache embeddings*

**for** $string$ in $\bigcup_k (\{r^{(k)}\} \cup \bigcup_{j=1}^{d_k} \{s_j^{(k)}\} \cup \bigcup_{j \in \mathcal{I}_{cat}^{(k)}} \{x_{ij}^{(k)}\}_{i=1}^{n_k})$ **do**

    $\bar{string} \leftarrow f_{LLM}(string)$

    Add $string : \bar{string}$ to $\bar{\mathcal{D}}$

**end for**

*# Training*

**repeat**

    Sample dataset index $k$ and sample batch of size bs without replacement.

    Load $(\bar{r}^{(k)}, \bar{m}^{(k)}, \bar{s}_1^{(k)}, ..., \bar{s}_{d_k}^{(k)}, \bar{X}_{cat}, X_{num}, \bar{C}^{(k)}$ (we ignore index $k$ from here)

    Sample $t \sim p_T(t)$

    Sample $\epsilon_{num} \in \mathbb{R}^{\text{bs} \times (d_k - |\mathcal{I}_{cat}^{(k)}|) \times d_h}$ with $\epsilon_{jl} \overset{iid}{\sim} N(0,1)$

    Sample $\epsilon_{cat} \in \mathbb{R}^{\text{bs} \times |\mathcal{I}_{cat}^{(k)}| \times d_h}$ with $\epsilon_{jl} \overset{iid}{\sim} N(0,1)$

    *# Add noise to data*:

    $\tilde{X}_{num} \leftarrow$ NumSched.add_noise$(X_{num}, \epsilon_{num}, t)$

    $\tilde{X}_{cat} \leftarrow$ CatSched.add_noise$(X_{cat}, \epsilon_{cat}, t)$

    *# Forward pass*:

    $\tilde{t} \leftarrow g_t(\text{PosEmb}(t))$

    $\tilde{m} \leftarrow g_r(\bar{m})$

    $\tilde{s}_j \leftarrow g_s(\bar{s}_j), \forall j$

    $h_j \leftarrow g_{cat}^{in}(\tilde{x}_j), \forall j \in \mathcal{I}^{(k)}$

    $h_j \leftarrow g_{num}^{in}(\tilde{h}_j), \forall j \notin \mathcal{I}_{cat}^{(k)}$

    $h_j \leftarrow \tilde{t} + \tilde{r} + \tilde{s}_j + h_j, \forall j$

    $\tilde{H} \leftarrow g_T(H)$

    $\tilde{x}_j \leftarrow g_{cat}^{out}(\tilde{h}_j), \forall j \in \mathcal{I}^{(k)}$

    $\hat{x}_j \leftarrow g_{num}^{out}(\tilde{h}_j), \forall j \notin \mathcal{I}_{cat}^{(k)}$

    *# Get probabilities for categoricals*:

    $\hat{x}_j \leftarrow \text{softmax}(g_f(\bar{C})\tilde{x}_j), \forall j \in \mathcal{I}_{cat}^{(k)}$ (Eq. 1)

    *# Compute loss and backwards, Eq. 2:*

    $\mathcal{L}_{cat} =$ CatSched.loss$(X_{cat}, \hat{X}_{cat}, t)$

    $\mathcal{L}_{num} =$ NumSched.loss$(X_{num}, \hat{X}_{num}, t)$

    Update $\Delta\Theta \propto -\nabla_\Theta \frac{1}{d_k}(\mathcal{L}_{num} + \mathcal{L}_{cat})$

**until** convergence

---

**Algorithm 2:** Generation phase. For diffusion schedulers, we use CDCD (Dieleman et al., 2022) for categoricals and VESDE Song et al. (2020) for numericals, denoted by respectively "CatSched" and "NumSched", which have the same timestep scheduling (denoted by Sched.timesteps($T$)). Capitals $X$ and $H$ denote the tensors of dimensions (batch size, number of columns, hidden dimension)

---

**Input:** metadata, $\bar{m}, (\bar{s}_j)_j, \bar{C}, \mathcal{I}_{cat}^{(k)}$, diffusion noise schedulers CatSched and NumSched, together denoted as "Sched", transformer $g_T$, primary networks $g_f, g_s, g_r, g_t, g_{cat}^{in}, g_{num}^{in}, g_{cat}^{out}, g_{num}^{out}$.

Sample $\tilde{X}_T \overset{iid}{\sim} N(0, I)$.

$\tilde{m} \leftarrow g_r(\bar{m})$

$\tilde{s}_j \leftarrow g_s(\bar{s}_j), \forall j$

**for** $t$ in Sched.timesteps($T$) **do**

  *# Single diffusion step*:

  $\tilde{t} \leftarrow g_t(\text{PosEmb}(t))$

  $h_j \leftarrow g_{cat}^{in}(\tilde{x}_{j,t}), \forall j \in \mathcal{I}^{(k)}$

  $h_j \leftarrow g_{num}^{in}(\tilde{x}_{j,t}), \forall j \notin \mathcal{I}_{cat}^{(k)}$

  $h_j \leftarrow \tilde{t} + \tilde{r} + \tilde{s}_j + h_j, \forall j$

  $\tilde{H} \leftarrow g_T(H)$

  $\tilde{x}_j \leftarrow g_{cat}^{out}(\tilde{h}_j), \forall j \in \mathcal{I}^{(k)}$

  $\hat{x}_{j,t-1} \leftarrow g_{num}^{out}(\tilde{h}_j), \forall j \notin \mathcal{I}_{cat}^{(k)}$

  $\hat{x}_{j,t-1} \leftarrow \text{softmax}(g_f(\bar{C})\tilde{x}_j), \forall j \in \mathcal{I}_{cat}^{(k)}$ (Eq. 1)

  *# Take diffusion step*:

  $\tilde{X}_{cat,t-1} = \text{CatSched.step}(\tilde{X}_{cat,t}, \hat{X}_{cat,t-1}, t)$

  $\tilde{X}_{num,t-1} = \text{NumSched.step}(\tilde{X}_{num,t}, \hat{X}_{num,t-1}, t)$

**end for**

Output: $\tilde{X}_0$

---