# OpenReview forum: "LATABLE: TOWARDS LARGE TABULAR MODELS"
_ICLR.cc/2025/Conference — Submitted to ICLR 2025_

### Official Review · Reviewer_YeUJ · 2024-10-31

**Soundness:** 2
**Presentation:** 1
**Contribution:** 3
**Rating:** 3
**Confidence:** 3

**Summary:**

This work proposes LaTable, a Diffusion model for tabular data generation that can be pretrained on large collections of tabular datasets and used for unconditional and conditional generation on unseen datasets in a few-shot manner. The model is evaluated in a few-shot generation setup where it performs comparably against other techniques.

**Strengths:**

Strengths:

* The present work is one of the first works that trains a generative transformer model across several dataset. The collection of datasets used to train the model presented is bigger than of any other works I am aware of, thereby making a reasonable step towards scaling up tabular models.

* Evaluation through ablation studies : I like the evaluation done through several ablation studies, showing the usefulness of the individual components. The results support the respective design choices.

* The Limitation and Discussion Section is honest and insightful.

**Weaknesses:**

Weaknesses
* The write-up is quite hard to understand and the notation seems overwhelming at some points. I know the basics of denoising diffusion models, but I was not quite able to follow in what space the diffusion happens and how exactly it is mapped back and forth into the tabular representation. Figure 1 doesn’t help much to gain a better understanding. In particular, I would have appreciated a section which would detail the generation procedure. I do not understand where in Figure 1 noise is input to generate synthetic samples and in diffusion models there are different generation paradigms as well, such as latent diffusion etc. I do not think the write-up is very accessible in its current form.
* Related Work. This is not the first attempt to build a tabular foundation models trained on multiple datasets. Notable approaches include TabPFN (Hollmann et al., 2023). I also wonder how LaTable compares to other approaches (although mainly focused on classification), such as Yak et al. (2023) or Zhu et al. (2023). It is unfortunate, that these competing approaches are neither discussed nor compared in the evaluation.
* Conditional Generation / Classification is not evaluated. The authors describe how conditional generation can be implemented with LaTable, but do not test is as far as I see. This also trivally allows to use LaTable for the classification task (by conditioning on all tabular features and letting the model generate the label). Here, it would be insightful to compare LaTable’s performance to models such as TabPFN, XTab, or GREAT. Also in addition, zero-shot and fine-tuning with the entire dataset should be considered for a comprehensive evaluation.
* The evaluation only uses ML Efficiency (Train Synth., Test Real, TSTR). For a comprehensive picture, the performance of a model trained on the real data should be included as an upper baseline to assess the data quality gap in the TSTR table. In addition, there could be further data quality metrics, including metrics such as the Discriminator metric (e.g. used in Borisov et al., 2023) where a model is trained to differentiate between original and synthetic data and its performance is reported. Also some quantitative results could complement the evaluation.

The overall impression of the submission suggests that not much care was taken to prepare the current manuscript for submission and several important things have been neglected:
* The citation format is incorrect, citep is not used properly
* The Appendix and Supplementary materials are missing
* There are several formatting issues, e.g., Figure 2
* Text in formulas should be wrapped in \text, e.g. \text{softmax} (eqn. 1, 2)
* There is no code available.

While these points may be fixed, I think for submission at a respected venue such as ICLR more care should be taken.

**Summary.** Overall, the submission seems to be rushed and I think a thorough revision is needed. This should include a more accessible write-up, studying classification performance and additional data quality metrics, and comparing with other attempts for building large tabular models.


Typos:
* Please check if “e.g.” should be followed by a comma.
* l. 237: transformer‘s
* l. 207 mathbb{R}  instead of normal R
* Caption of Figure 2: Capitalization of LaTable

----------------

**References**

Zhu, B., Shi, X., Erickson, N., Li, M., Karypis, G., and Shoaran, M. XTab: cross-table pretraining for tabular transformers. In Proceedings of the 40th International Conference on Machine Learning (pp. 43181-43204), 2023

Yak, Scott, Yihe Dong, Javier Gonzalvo, and Sercan Arik. "IngesTables: Scalable and Efficient Training of LLM-Enabled Tabular Foundation Models." In NeurIPS 2023 Second Table Representation Learning Workshop. 2023.

Hollmann, Noah, Samuel Müller, Katharina Eggensperger, and Frank Hutter. "TabPFN: A Transformer That Solves Small Tabular Classification Problems in a Second." In The Eleventh International Conference on Learning Representations, ICLR 2023.

Vadim Borisov, Kathrin Sesler, Tobias Leemann, Martin Pawelczyk, and Gjergji Kasneci. Language models are realistic tabular data generators. ICLR 2023.

**Questions:**

Can you explain the generation procedure? Where is the noise inserted?

Can you differentiate LaTable from other large tabular models, e.g. TabPFN? How does it compare?

---

> ### Author Response · Authors · 2024-11-26
> **We thank you for your feedback**
>
> Dear Reviewer YeUJ, many thanks for your insightful review. We thank you for acknowledging and summarizing succinctly the strengths of this paper. The questions you raise are important, and we hope the community recognizes the significance of discussing these questions. Here are our responses to the points you have raised:
>
> Weaknesses:
> 1. Indeed, the notation is rather overwhelming and complex; even though this comes with the advanced setting (cross-dataset generation with tabular transformers and diffusion models), we agree the readability and clarity can be improved. In particular, we have included pseudo-code for training as well as generation in Appendix A.
> 2. You correctly note that Hollmann, Yak, and Zhu build tabular foundation models trained on multiple datasets, so do (Zhang et al, 2023) and (Dinh et al., 2022). However, we note that these approaches focus on supervised learning, and not generative learning. This implies we cannot compare directly to them, as our method does not provide predictions, and it is non-trivial to extend their method to generation. We believe the suggestion may stem from us using downstream model performance (i.e. TSTR a.k.a. Model Efficiency), which we could compare to their works. However, we note this comparison is not entirely valid: the TSTR metric is dependent on which downstream classifier architecture we use—finding the best classifier is beyond the scope of this work and we use the standard CatBoost model. Consequently, any generative model’s TSTR performance (including LaTable) could in practice be improved further, and comparing this to a supervised foundation model performance would not be entirely fair. To the best of our knowledge, the only generative tabular foundation model is currently GReaT.
> 3. Thank you for raising this. Although LaTable can be extended to conditional generation, the scope of this work is only unconditional generation. We agree that conditional generation is also promising for prediction. Future work could consider how conditional generation can be made efficient (i.e. conditional generation models $P(Y|X)$, whereas standard supervised learning algorithms are only interested in a single statistic $E(Y|X)$), and whether a generative approach to prediction can provide heuristic uncertainty (e.g. even for numericals we have access to $P(Y|X)$ and could thus approximate variance or other statistics).
> 4. Thank you for raising this. The discriminator metric used in (Borisov et al., 2023) we do not include: this is not very insightful for the few-shot setting. This metric achieves an almost perfect score (AUC>0.98) on both LaTable and baselines, by finding simple generation artifacts. These generation artifacts are deemed to occur, since the generative models have only seen a small number of real samples, and cannot therefore be expected to approximate the distribution perfectly. On the other hand, the discriminator used in the metric is trained on the real test set and synthetic sets (each a few thousand), and will be able to find and exploit artifacts easily. The metric could be changed to a simpler model to make it more fair. For example, we tested a linear regression model (with categoricals one-hot encoded), for which we found discriminator performance: 0.64, 0.62, 0.63 for TVAE, GREAT, and LaTable, respectively. These numbers fall do not show significant differences, hence we unfortunately find these not insightful enough for the paper.
>
> We hope this response has addressed your central concern. Please see our general comment for more details about the experiment setup. We will continue to expand on the limitations outlined in the paper.

---

### Official Review · Reviewer_SWVR · 2024-11-03

**Soundness:** 2
**Presentation:** 2
**Contribution:** 2
**Rating:** 3
**Confidence:** 2

**Summary:**

The paper introduces "LaTable," a novel generative model for tabular data based on diffusion techniques, specifically aimed at large-scale tabular data generation. LaTable displays early signs of "scaling laws" observed in foundation models. The authors empirically demonstrate that their model outperforms existing generative models in ood settings.

**Strengths:**

- The paper addresses the underexplored domain of large-scale tabular data modeling, a departure from traditional focus areas in foundation models such as text and vision.
- The model meets several carefully formulated desiderata: cross-dataset generation, mixed-type handling, use of textual metadata, and equivariance to column order. The authors thoroughly answer each desiderata in their model design.
- LaTable represents an important step toward creating generative models that can be applied to diverse tabular datasets.

**Weaknesses:**

I mostly found it hard to understand the architecture of the model and the training objectives you used on it. I am from outside the tabular data community so this may be the reason, but I think that it should be clear to people outside the community as well. It is clear that you very carefully designed the architecture to meet all your requirements but I wasn't sure in the end what is the input/output, how you train everything end-to-end. A more higher level description is required instead of diving in directly to satisfying desiderata.

**Questions:**

None

---

> ### Author Response · Authors · 2024-11-26
> **We thank you for your feedback**
>
> Dear Reviewer SWVR, many thanks for your constructive feedback. We thank you for clearly raising your confusions and concerns when reading the paper. Indeed, the notation is rather overwhelming and complex; even though this comes with the advanced setting (cross-dataset generation with tabular transformers and diffusion models), we agree the readability and clarity can be improved. In particular, we have included pseudo-code for training as well as generation in Appendix A. We hope this response has addressed your central concern. Please see our general comment for more details about the experiment setup. We will continue to expand on the limitations outlined in the paper.

---

### Official Review · Reviewer_pV4v · 2024-11-03

**Soundness:** 2
**Presentation:** 2
**Contribution:** 2
**Rating:** 5
**Confidence:** 3

**Summary:**

The work introduces a generative model LaTable designed specifically for tabular data generation. Recognizing the ubiquity and challenges of tabular data, the authors focus on managing heterogeneous tabular data features, such as categorical and numerical variables, across various datasets. LaTable leverages metadata to improve the generation process and is built upon an encoder-only transformer. The model is evaluated through few-shot learning scenarios to validate its cross-dataset generalization capabilities. Additionally, LaTable shows early signs of scaling laws, commonly observed in foundation models in other domains.

**Strengths:**

The paper clearly outlines four primary design goals—cross-dataset generation, handling of categorical and numerical features, use of textual context, and column order equivariance—and effectively aligns these with specific model design choices. Additionally, it identifies scaling laws unique to tabular data, which is valuable given that this area has not been thoroughly explored within the scope of tabular foundation models.

**Weaknesses:**

1. LaTable shows limited robustness on non-binary classification tasks, such as multi-class classification and regression, suggesting constrained generalization across different task types.
2. The descriptions of datasets and baseline models are brief and lack detail.
3. The evaluation metrics are limited, primarily focusing on downstream performance.
4. Figure 2 is oversized.
5. Although the paper acknowledges issues of data bias and fairness, it does not explore practical approaches to detecting or mitigating these biases in real-world applications.

**Questions:**

1. Will you consider including additional metrics beyond downstream performance, such as low-order and high-order statistics [1]?
2. Will you consider adding more recent baselines, such as TABSYN [1] and TabDDPM [2]?
3. Figure 2 illustrates the effects of training set size and model parameters but does not address the impact of the number of categories and numerical features. Will you consider investigating this aspect?
4. Do you plan to release the codes?

[1] Zhang, Hengrui, et al. "Mixed-type tabular data synthesis with score-based diffusion in latent space." arXiv preprint arXiv:2310.09656 (2023).
[2] Kotelnikov, Akim, et al. "Tabddpm: Modelling tabular data with diffusion models." International Conference on Machine Learning. PMLR, 2023.

---

> ### Author Response · Authors · 2024-11-26
> **We thank you for your feedback**
>
> Dear Reviewer pV4v, many thanks for your constructive feedback. We thank you for clearly raising your confusions and concerns when reading the paper. Here are our responses to the questions you have raised:
>
> We clarify that TabDDPM is one of our baselines, and TabDDPM achieves good performance in our experiments. We plan to release the codes upon paper acceptance. We thank you for raising questions about other synthetic data metrics. Aside from your suggestions, additional metrics for synthetic data include coverage, density, and the Discriminator metric suggested by another reviewer. We remark that there is a reason that we don't include these metrics, namely that they are not very insightful for the few-shot setting. For example, the Discriminator metric used in (Borisov et al., 2023) achieves an almost perfect score (AUC>0.98) on both LaTable and baselines, by finding simple generation artifacts. These generation artifacts are deemed to occur, since the generative models have only seen a small number of real samples, and cannot therefore be expected to approximate the distribution perfectly. On the other hand, the discriminator used in the metric is trained on the real dataset and synthetic sets (each a few thousand), and will be able to find and exploit artifacts easily. The metric could be changed to a simpler model to make it more fair. For example, we tested a linear regression model (with categoricals one-hot encoded), for which we found discriminator performance: 0.64, 0.62, 0.63 for TVAE, GREAT, and LaTable, respectively. These numbers fall do not show significant differences, hence we unfortunately find these not insightful enough for the paper.
>
> We hope this response has addressed your central concern. Please see our general comment for more details about the experiment setup. We will continue to expand on the limitations outlined in the paper.

---

> > ### Comment · Reviewer_pV4v · 2024-12-02
> >
> > Thank you for your clarification! While I agree with the Reviewer avjV "the scope is limited", and my concerns have only partially been addressed. Therefore, I will maintain my score.

---

### Official Review · Reviewer_avjV · 2024-11-04

**Soundness:** 1
**Presentation:** 1
**Contribution:** 2
**Rating:** 3
**Confidence:** 3

**Summary:**

The paper introduces LaTable, a diffusion-based generative model for table generation. LaTable is designed to support flexible generation with varying numbers and types of features, utilizes additional context like dataset descriptions and feature information, and demonstrates equivariance with respect to column order. The authors conducted experiments showing that model performance scales with both model and table sizes, and that LaTable outperforms several baselines. Results also indicate that training across multiple datasets, incorporating textual context, and exposing the model to more data samples (e.g., table rows) or increasing the model size enhance its performance.

**Strengths:**

- The authors provide a clear motivation for the need for tabular generative models and present a model designed to meet the specific requirements of tabular data generation. The related works section is well-written, highlighting why LLM-based approaches are not optimal compared to their diffusion-based approach.
- The authors conducted comprehensive experiments, examining critical factors beyond general model performance, such as scalability, cross-dataset training procedures, and the impact of including textual context.

**Weaknesses:**

- The evaluation setup is unclear. The authors mention Cardio, URL, WiDS, Insurance, and Heloc as test datasets, citing Stoian et al. However, only URL, WiDS, and Heloc are covered in that paper; details on Cardio and Insurance datasets are not disclosed, and relevant citations for these datasets are missing. Additionally, the authors do not provide clear references to the baselines (e.g., it is unclear which papers CTGAN, TVAE, ARF, DDPM, and GREAT correspond to in L343).
- The authors state that all methods are fine-tuned on the test dataset (L348) but then fit a CatBoost model on the generated data from models fine-tuned on the test set (L346) to predict table elements from the test set itself (L347). Is this interpretation correct? If so, the metrics’ significance is unclear. For instance, a model that generates only the exact data it was fine-tuned on would make the generated synthetic training set overlap with the test set, which would improve CatBoost performance, but the metrics would not indicate the model’s ability to generate novel synthetic data instead of memorized training data.
- The paper's presentation is poor and requires improvement. References are incorrectly formatted (e.g., missing parentheses), figures exceed boundary limits (e.g., Figure 2), and the phrase “scaling law” is mentioned 14 times without citing a single scaling law paper.
- No results are provided for multiclass classification or regression tasks. Although some discussion is included in L467, quantitative comparisons on other tasks between the authors’ method and other baselines are necessary to showcase the general capability and utility of LaTable.

**Questions:**

Can the authors clarify what WhereIsAI/UAE-Large-V1 refers to in L171?

---

> ### Author Response · Authors · 2024-11-26
> **We thank you for your feedback**
>
> Dear Reviewer avjV, many thanks for your constructive feedback. We thank you for clearly raising your confusions and concerns when reading the paper. Here are our responses to the central question you have raised:
>
> We wish to clarify one key misunderstanding about our experiment setup, namely that we operate in the few-shot setting. Empirically, we notice that asking the model to generate synthetic data without any real data samples is unrealistic. Therefore, we make a weaker assumption, and assume that our application scenario is in the low-data regime. We show the model a very small portion of the calibration data, which is emphasized on both lines 300-308 as well as line 347 on page 6 of the manuscript. The whole picture of our experiment procedure is summarized as follows:
> 1. We begin with a tabular model trained to completion and a calibration dataset. We will use the dataset Adult to illustrate our point.
> 2. We perform minor fine-tuning of the tabular model on 100 data samples from Adult (which is very small for synthetic data generation models), and we generate synthetic data given the metadata from Adult.
> 3. We fit a well-known classifier model to the generated synthetic data, and evaluate on real data using downstream performance metrics such as accuracy, f1 score, and rocauc. We use the CatBoost classifier in this paper.
>
> We hope that we have clarified the experiment setup. The metrics’ significance should be clear now. Even though the model has seen a few samples of real data, memorizing those data will not help the model generate the entire calibration dataset. Understanding about the dataset is still required from the model because the generated synthetic dataset is orders of magnitude larger than the few-shot data.
>
> Regarding your question, "WhereIsAI/UAE-Large-V1" refers to the Universal Angle embedding model [1] (also shown in the footnote of the paper). We hope this response has addressed your central concern. Please see our general comment for more details about the experiment setup. We will continue to expand on the limitations outlined in the paper.
>
> [1] https://huggingface.co/WhereIsAI/UAE-Large-V1

---

> > ### Comment · Reviewer_avjV · 2024-12-01
> >
> > Thanks for the clarification! The authors have addressed my concern on the experiment setup, but the presentation/clarity issue still exists. Additionally, this work is limited to binary classification tasks and thus the scope is limited. Given the authors clarified the experiment setup, I raised my score to 3. I encourage the authors to improve the presentation of the manuscript and the scope of the work in future revisions.

---

### Author Response · Authors · 2024-11-26
**General Comment from the authors**

Dear reviewers, many thanks for your insights!

—Evaluating tabular foundation models via few shot synthetic data generation—

Seeing that a few reviewers are puzzled about the evaluation setting in this paper, we wish to emphasize lines 300-308 on page 6 of the manuscript. In this paper, we choose to benchmark tabular generative foundation models by evaluating its ability to generate synthetic data few-shot. We use “Train on Synthetic, Test on Real” (TSTR) downstream performance, also known as “Model Efficiency” (ME):
1. Take a LaTable model trained to completion and select a calibration dataset (e.g. Adult)
2. Perform minor fine-tuning of the generative model on 100 data samples from the target dataset. Note that this is very small for synthetic data generation models.
3. Generate synthetic data given the metadata from the target data.
4. We train a prototypical classifier $f$ (i.e. CatBoost in this paper, following TabDDPM) on the generated synthetic data.
5. We evaluate $f$ on real data using downstream performance metrics (i.e. AUC in this paper).

—Evaluation setting and related works—

Having clarified the experiment setting, we believe it is also insightful to discuss the related works. Many papers have studies tabular foundation models before us, including (Hollmann et al., 2023), (Yak et al., 2023), (Zhu et al., 2023), as well as (Zhang et al., 2023) and (Dinh et al., 2022). However, we note that these approaches focus on supervised learning, and not generative learning. This implies we cannot compare directly to them, as our method does not provide predictions, and it is non-trivial to extend their method to generation. We believe the suggestion may stem from us using downstream model performance (i.e. TSTR a.k.a. Model Efficiency), which we could compare to their works. However, we note this comparison is not entirely valid: the TSTR metric is dependent on which downstream classifier architecture we use—finding the best classifier is beyond the scope of this work and we use the standard CatBoost model. Consequently, any generative model’s TSTR performance (including LaTable) could in practice be improved further, and comparing this to a supervised foundation model performance would not be entirely fair. To the best of our knowledge, the only generative tabular foundation model is currently GReaT.

---

### Meta-Review · Area_Chair_pjGE · 2024-12-23

**Metareview:**

This paper introduces LaTable, a new diffusion-based generative model designed for tabular data that can be trained across multiple datasets. The authors present LaTable as one of the first attempts at building a tabular foundation model, addressing key challenges like handling heterogeneous feature spaces, incorporating metadata, and maintaining column order equivariance. The work demonstrates early signs of scaling laws similar to those observed in other foundation model domains, and shows superior performance compared to baselines in few-shot out-of-distribution data generation scenarios.

The reviewers unanimously provided negative scores and raised several significant concerns about the the paper’s presentation and scope. The reviewers seemed to be generally confused about the model architecture and training objectives. Besides, reviewers pointed out limited evaluation and lack of comparison with other recent tabular foundation models (though authors note these focus on supervised rather than generative learning). While reviewers acknowledge the paper's novel contributions toward scalable tabular generation, they recommend rejection due to the need for substantial improvements in presentation quality, broader experimental evaluation beyond binary tasks, and more comprehensive comparison with related work. The authors' rebuttal clarifies the few-shot evaluation setup and reasoning behind metric choices, but reviewers maintain their assessment that the current manuscript requires significant revision to meet publication standards.

**Additional Comments On Reviewer Discussion:**

See above.

---

### Decision · Program_Chairs · 2025-01-22

Reject